# The Role of Hypoxia and Cancer Stem Cells in Development of Glioblastoma

**DOI:** 10.3390/cancers15092613

**Published:** 2023-05-04

**Authors:** Tingyu Shi, Jun Zhu, Xiang Zhang, Xinggang Mao

**Affiliations:** 1Department of Neurosurgery, Xijing Hospital, Fourth Military Medical University, Xi’an 710032, China; 2Tangdu Hospital, Fourth Military Medical University, Xi’an 710024, China; 3State Key Laboratory of Cancer Biology, Institute of Digestive Diseases, Xijing Hospital, The Fourth Military Medical University, Xi’an 710032, China

**Keywords:** glioblastoma, stem cell, hypoxia, niche, autophagy, therapy

## Abstract

**Simple Summary:**

As one of most malignant tumors in brain, glioblastoma (GBM) is lack of effective treatment and the prognosis of GBM patients is still very poor despite accumulated progresses. Hypoxia is an essential factor for the initiation and progression of GBM, especially for the glioma stem like cells (GSCs). Hypoxia induced many target genes which form a complicated molecular interacting network, influencing a lot of tumor behaviors by regulating key signal pathways. In addition, hypoxia has great impact on the interplayed niches of GCSs. Here, by systematically reviewing the role of hypoxia on the maintenance of GSCs and the development of GBM, and analyzing the related molecular mechanisms, we integrated the hypoxia related tumor features of GBM. This summary helps to deepen our knowledge of the tumorigenic mechanisms of GBM, and can help to develop novel therapeutic strategies targeting hypoxia to improve the survival of GBM patients.

**Abstract:**

Glioblastoma multiform (GBM) is recognized as the most malignant brain tumor with a high level of hypoxia, containing a small population of glioblastoma stem like cells (GSCs). These GSCs have the capacity of self-renewal, proliferation, invasion and recapitulating the parent tumor, and are major causes of radio-and chemoresistance of GBM. Upregulated expression of hypoxia inducible factors (HIFs) in hypoxia fundamentally contributes to maintenance and progression of GSCs. Therefore, we thoroughly reviewed the currently acknowledged roles of hypoxia-associated GSCs in development of GBM. In detail, we recapitulated general features of GBM, especially GSC-related features, and delineated essential responses resulted from interactions between GSC and hypoxia, including hypoxia-induced signatures, genes and pathways, and hypoxia-regulated metabolic alterations. Five hypothesized GSC niches are discussed and integrated into one comprehensive concept: hypoxic peri-arteriolar niche of GSCs. Autophagy, another protective mechanism against chemotherapy, is also closely related to hypoxia and is a potential therapeutic target for GBM. In addition, potential causes of therapeutic resistance (chemo-, radio-, surgical-, immuno-), and chemotherapeutic agents which can improve the therapeutic effects of chemo-, radio-, or immunotherapy are introduced and discussed. At last, as a potential approach to reverse the hypoxic microenvironment in GBM, hyperbaric oxygen therapy (HBOT) might be an adjuvant therapy to chemo-and radiotherapy after surgery. In conclusion, we focus on demonstrating the important role of hypoxia on development of GBM, especially by affecting the function of GSCs. Important advantages have been made to understand the complicated responses induced by hypoxia in GBM. Further exploration of targeting hypoxia and GSCs can help to develop novel therapeutic strategies to improve the survival of GBM patients.

## 1. Introduction

Glioblastoma (GBM) is identified as one of the most malignant solid brain cancers, with an average annual incidence of about 4.45 per 100,000 population [1]. GBM patients present clinical manifestations of headache, weakness, vague vision, seizure and/or dizziness, depending on tumor location and degree of neurological impairment. The average diagnostic age of GBM patients is 64-years-old, with more females than males (6:4) [2]. With advancement of technology, novel imaging tools offer great assistance for oncologists in the diagnosis of GBM. Apart from conventional computed tomography (CT) and magnetic resonance imaging (MRI), dynamic contrast-enhanced (DCE) MRI, functional MRI, diffusion tensor imaging (DTI), magnetic resonance spectroscopy (MRS), diffusion-weighted imaging (DWI), single-photon emission computed tomography (SPECT), and positron emission tomography (PET) are all useful in clinical practice [3,4].

It has been proposed that GBMs have intrinsic cellular heterogeneity which consists of differentiated cells, quiescent cells, and glioblastoma stem cells (GSCs) [5]. Differentiated cells mainly contribute to tumorigenesis, proliferation, and invasion of glioblastoma. Quiescent cells are able to transdifferentiate into stem-like cells, and re-acquire self-renewal ability [6]. GSCs are recognized as reservoirs of tumor-initiating cells, accounting for therapeutic failure and GBM recurrence. The hypothesis of cancer stem cell (CSC) arises from human acute leukemia. Hematopoietic stem cells (HSCs) are a small population of multipotent cells with the potential of proliferation, self-renewal, differentiation, and regeneration of original tumors [7]. GSCs kept a certain degree of neural stem cell (NSC) features such as self-renewal and multi-differentiation potential [8]. NSC was first described in grown-up mammalians and mainly exists in two regions: one is the subventricular zone (SVZ) between the striatum and the lateral ventricle, and another is the subgranular zone (SGZ) within the dentate gyrus of the hippocampus [9,10].

Hypoxia plays a paramount important role in neuronal development which is a prerequisite for the neural crest cell migration [11]. Hypoxia mainly functions through hypoxia-inducible factors (HIFs, HIF1α, and HIF2α). In normoxia, HIF1α is hydroxylated and combined with a cancer suppressor Von Hippel-Lindau (VHL) to undergo ubiquitination process [12]. Under hypoxia, the HIF1α protein is speedily accumulated within cells and contributes to subsequent gene transactivation. HIF1α promotes glycolysis via upregulating critical enzymes of glycolytic pathway, such as HK2 (hexokinase 2) and pyruvate PDK1 (pyruvate dehydrogenase kinase 1). The literature has reported that HIF1α regulates stemness and differentiation of early NSC population via activating neural repressor Hes1 [13]. Suppression of HIF1α by meloxicam could exert antiproliferative efficacy in hepatocellular carcinoma (HCC) and lead to caspase-reliant apoptosis of HCC in hypoxia [14]. HIF2α is an intimate isoform of HIF1α [15]. Contrary to widespread expression of HIF1α in nearly all cells, HIF2α is selectively expressed in stem cells and endothelial cells of cancer. HIF1α shows high sensitivity towards oxygen concentration while HIF1β demonstrates constitutive expression regardless of oxygen concentration [16]. The HIF1α/HIF1β compound could translocate into the nucleus and further command genes that contain the hypoxia-response consensus sequence (HRE) [17].

Hypoxic areas in GBMs could be attributed to multiple factors such as upregulated cellular proliferation, insufficient oxygen diffusion, widespread tissue necrosis, broken blood-brain barrier, and aberrant tumor vascularization. Hypoxia is closely linked with the neoplastic biology of GBMs. Upregulated HIF expression in hypoxia promotes proliferation, infiltration, and self-renewal of GSC, ultimately leading to an enhanced level of therapeutic-resistance. However, the relationship between hypoxia and GSCs in the development of GBM is not clearly elaborated. Therefore, in our review, we recapitulate general features of GBM, describe GSC-related features, and delineate interactions between GSC and hypoxia. Given the importance of hypoxia for the initiation and progression of GBMs and GSCs, comprehensive study and discussion of these issues would give us more insights into the biological features of GBMs and provide novel avenues to develop promising treatments for GBMs targeting hypoxia and GSCs.

## 2. GSC and Hypoxia-Related Signatures

One of the central issues for studying GSCs is to identify the GSCs, primarily by using suitable molecular markers. It has been reported that CD9, CD133 (prominin-1), Olig2, integrin αβ, aldehyde dehydrogenase (ALDH), CD44, Sox2, Oct4, nestin, and the feature of side population (SP) can be used as signatures of GSCs [17,18,19,20,21] (Table 1). These markers are useful in the identification of stem cells, thus propelling relative studies. Via detecting expression of Oct4, Olig2, and nestin, it was reported that ING5 (a member of the epigenetic regulators ING family) could accelerate self-renewal of GSC, enhance its stem-cell pool, and block its lineage differentiation [22]. When hypoxia is presented, many of these GSC markers are upregulated. Seidle et al. observed that SP-related genes are upregulated in hypoxia in three adherent glioma cells [23].They also discovered that SP marker genes are highly expressed in both peri-vascular and hypoxic niches where both HIF1α and HIF2α are highly expressed [23]. However, whether all these cell markers can be applied precisely to identify stem cells remains controversial. Some CD133^−^ cells also have the properties of GSCs and high plasticity of generating CD133^+^ cells. Currently, the gold criterion to determine GSCs remains the competence of recapitulating original parent tumors under the condition of orthotopical transplantation. Therefore, further investigations are necessary to uncover intrinsic GSC features.

## 3. GSC and Hypoxia-Related Genes

GBMs can be further classified into four subtypes: mesenchymal, neural, proneural, and classical subtypes. Neurofibromin 1 (NF1) deletion, chromosome 7 enrichment, platelet-derived growth factor (PDGF) amplification, and tumor suppressor PTEN deficiency are discovered in these four types respectively [24]. In addition, p16 loss, epidermal growth factor receptor (EGFR) amplification, chromosome 22q loss, TP53 mutation, and CDKN2A loss are the most common and prominent signal alterations in GBMs [25] (Table 1). Among them, TP53 mutation is present in both primary and recurrent GBMs [26]. Recurrent GSC is able to accumulate temozolomide-associated mutations over primary GSC after chemo-therapy [26]. IDH1 is an oncogene which localizes in cytoplasm and peroxisome. IDH1 mutation is a symbol of early tumorigenesis, suppression of which could enhance sensitivity of GBM to chemotherapy. Glioma patients with IDH-mutation display better prognoses compared to those with IDH-wildtype. In the latest WHO classification, GBM only represents IDH-wildtype GBM, while IDH-mutant GBM, which was considered to account for 10% of GBMs in the past [27], is considered as a different subtype of diffuse glioma [28].

Considering the intimate relationship between GBMs and hypoxia, deep insight into genetic alterations of GBMs under hypoxic conditions is essential. Evidence has revealed that there is an intimate interplay between IDH1/2 and HIFs. Mutated IDH1/2 leads to elevated expression of oncometabolite R-2-hydroxyglutarate (2HG), which then decreases HIF1α and HIF2α levels [29]. Interestingly, IDH mutation alone inhibits tumor growth while the combinatory effect of IDH1/2 and HIF promotes neoplastic growth, contributing to unfavorable prognosis in GBM patients. It is reported that expression of monocarboxylate transporter-4 (MCT4), protein phosphatase 2A (PP2A), Krupple-like 4 (Klf4), and ATP-binding cassette B1 (ABCB1) are upregulated under hypoxic conditions and lead to shorter survival spans of GBM patients [30,31,32,33]. It is acknowledged that HIF1α level increases after mammalian target of rapamycin (mTOR) is dysregulated [34]. Multiple genes participate in the above pathways, such as PTEN (phosphatase and tensin homolog), PML (promyelocytic leukemia), and EGFR [35,36].

## 4. GSC and Hypoxia-Related Pathways

Accumulated research showed that numerous signaling pathways are altered in GBM, such as PI3K/AKT/mTOR, MAPK, STAT3/bcl2, PI3K/RhoA/C, HIF/IDH1/2, VEGF, EGF, Wnt/βcatenin, and Notch [37,38,39,40] (Table 1). It has been indicated that ING5 (a member of the epigenetic regulators ING family) could increase the activity of PI3K/AKT via facilitating transcription of the calcium channel as well as the follicle stimulating hormone signaling gene, to maintain self-renewal of GSCs which partially causes resistance and recurrence of GBMs [22].

Under hypoxia, there are several alterations in GBM-related pathways. Grassi et al. reported that hypoxia could induce upregulation of Delta like non-canonical Notch ligand 1 (DLK1) in GBM, thus promoting colony formation of GBMs as well as gene expression of GSC markers [41]. A recent study showed that CBF1, a cardinal transcriptional modulator of Notch signaling pathway, could be activated by hypoxia to promote proliferation of GSCs and accelerate epithelial-to-mesenchyme transition (EMT), further enhancing chemoresistance of GBMs [42].

Vascular endothelial growth factor (VEGF) is a pro-angiogenic factor that mediates vascular permeability and angioedema. VEGFR2 is a receptor of VEGF. In hypoxia, both VEGF and VEGFR2 are up-regulated by HIF1α and overexpressed in GBM, accelerating tumor progression and invasion [43]. In addition, enhanced HIF1α expression could activate the JAK1//2-STAT3 pathway that closely associates with VEGF secretion, thus promoting self-renewal of GSCs [44]. Bevacizumab is a kind of anti-VEGF monoclonal antibody currently used as a second-line agent which shows efficiency in decreasing aberrant vascularization of GBM [45]. Brefeldin A is another inhibitor of VEGF in GBM [46]. However, anti-VEGF treatments inevitably lead to therapy resistance. An investigation indicated that resistance to anti-VEGF therapy in GBM is facilitated by elevation of regulatory T-cell (Treg), which might serve as potential targets with both immunologic and anti-VEGF effects [47].

The Wnt pathway has an intimate association with GSC features, being able to reduce CD133 and Nestin under both aerobic and anaerobic conditions. In hypoxia, HIF1α upregulates expression of both TCF-1 and LEF-1 to cooperate with Wnt signaling in GBM, reprograming GSC phenotype towards a more differentiated and less aggressive one. A study has revealed that hypoxia-induced Wnt activation could inhibit Notch pathway in primary GBM and enhance chemosensitivity of GBM cells towards temozolomide (TMZ) therapy [48].

Transforming growth factor β (TGFβ) is a downstream gene of HIF1α with two isoforms TGFβ1/β2. TGFβ plays significant roles in GBM progression and recurrence [49], and can promote GSC invasion in vitro [50]. Integrin avβ8 is overexpressed in GSC and crucial for GSC self-renewal and GBM tumorigenesis. It was demonstrated that avβ8 integrin mediates GBM progression via promoting TGFβ1-induced DNA replication, thus the avβ8-integrin-TGFβ1 axis might function as a therapeutic target of GBM [51]. Dedifferentiation of non-stem cells into stem cells is known to be involved in EMT. Under hypoxia, GSCs could release TGFβ1 to promote EMT, leading to increased quantities of GSC and poor outcomes of GBM patients [49]. Interestingly, TGFβ is recognized as an upstream regulator of VEGF, and modulating VEGF and TGFβ signaling pathways collectively could effectively control neoplastic growth of GBMs [52].

Long noncoding RNA (lncRNA) H19 displays a tumorigenic role in GBMs under hypoxia. The research has highlighted that targeting lncRNA H19 might be a potential therapeutic strategy for GBMs [53]. Hypoxia-inducible and lipid droplet associated protein (HILPDA, also identified as HIG2) is inherently overexpressed in GBMs and enhanced by hypoxia, contributing to unfavorable prognosis of GBM patients [54].

## 5. GSC and Hypoxia-Related Metabolism

The “Warburg effect” refers to elevated levels of aerobic glycolysis in which pyruvate is transformed into lactate instead of entering Krebs cycle. Hypoxia could affect one-carbon metabolism of GSC via over-expression of aerobic glycolytic pathway enzymes such as LDHA (lactate dehydrogenase A), PFK1 (phosphofructokinase 1), and HK2, as well as down-regulation of vitamin B12 transporter protein TCN2 (Figure 1). TCN2 is indispensable in the process of GSC transformation into the highly malignant mesenchymal/CSC profile [55]. The IDH3α/cSHMT (cytosolic serine hydroxymethyltransferase) signaling axis is recognized as a novel regulatory target of one-carbon metabolism in GBM [56]. Epigenetic regulation via histone alteration, DNA methylation, and non-coding RNA could also mediate glycolytic metabolism in GBM [57]. Apart from glucose metabolism of GSCs, HIFs also play a vital role in the metabolism of amino acid. LAT1 is a transporter of branched-chain amino acid (BCAA) while BCAT1 is a metabolic enzyme of BCAA. It was reported that HIF1α and HIF2α increase both mRNA and protein levels of LAT1 and BCAT1 in GBM under hypoxia [15]. In hypoxic condition, the major carbon fuel of GBM cells partially convert from glucose to glutamine [58]. α-ketoglutarate (αKG) is a medium metabolite in tricarboxylic acid (TCA) cycle. αKG and the associated amino acid glutamate are two key factors of GBM metabolic alterations [59]. It has been identified that Acyl-CoA-Binding protein could facilitate tumorigenesis of GBMs via promoting fatty acid oxidation [60]. In conclusion, hypoxia-related metabolism is essential for the initiation and progression of GBM, which is involved in complicated processes and deserves further investigation to obtain more insights of the GSC properties.

## 6. GSC and Hypoxia-Related Vasculature

Vasculature plays a crucial role in GBM initiation and progression. There exist five potential mechanisms of GBM vascularization: angiogenesis, vasculogenesis, vessel mimicry, vessel co-option, and intussusception [61] (Figure 2). Angiogenesis in GBM could be attributed, to a large extent, to interplay between GSCs and endothelial cells via VEGFR, Notch, DLL-4, and nitric oxide (NO) signals. Recruitment of the bone marrow-originated endothelial progenitor cell (EPC) by SDF-1/CXCR-4 is the precondition of vasculogenesis [62]. During vascular mimicry, tumor cells constitute vascular channels which have no endothelial cells but are still able to transport erythrocytes. Proliferation of GBM was initiated by vessel co-option, followed by angiogenesis once tumor mass grew to a certain volume [63]. Intussusception is a type of vascular remodeling where a blood vessel divides into two. In vasculatures of GBMs, GSCs interact closely with adjacent cells. ECs release NO that diffuses into GSCs. GSCs generate pro-angiogenic VEGF-A to facilitate growth of ECs [64]. It has been testified that the PAX6/DLX5-WNT5A pathway might be an underlying regulator of interaction between GSC and EC in GBM [65]. Interplay between GSC and EC is important for GBM progression, which is affected by hypoxia. In spite of obvious vascularization, the microenvironment of a GBM is usually hypoxic which might be attributed to tortuous, poorly-organized, and insufficiently-perfused tumor vessels. Hypoxia increased expression of extracellular adenosine that activates the A_3_ adenosine receptor (A_3_ AR) to promote trans-differentiation of GSC into EC [66]. Hypoxia also induces fusion of GSC with EC through pseudo-endothelialization. Astrocytes mainly produce extracellular matrix (ECM) proteins such as proteoglycan, collagen, and laminin [67] (Figure 3). As important components of ECM, integrin α6, the receptor for laminin, is enriched in GSC, and promotes interactions between GSC and EC [68,69].

## 7. GSC and Hypoxia-Related Niches

Conventional therapies mainly target CSCs, but to some extent, surrounding niches also contribute to the malignancy of neoplasm [70]. The tumor microenvironment (TME) of GBM has attracted more attention in recent years, which generally includes dendritic cells, fibroblasts, vessels, macrophages, and cancer-draining lymph nodes. TME might assist self-renewal and stemness of GSCs, acting as novel therapeutic targets. Five niches are recognized and elucidated in detail in the current paper: peri-vascular niche, immune niche, hypoxia/necrotic niche, ECM niche, and peri-arteriolar niche [71]. The TME may contain more cell types that have impacts on GSCs. For example, astrocytes express Sonic hedgehog (SHH) and modulate the self-renewal of GSC and progression of GBM [72].

### 7.1. Peri-Vascular Niche of GSC

There is bidirectional crosstalk between a GSC and molecules from its peri-vascular niche such as endothelial cells (ECs), pericytes, astrocytes, and microglia/macrophages. The Notch ligand of EC cooperates with the Notch receptor of GSC to activate the Notch pathway, further promoting self-renewal of GSC [73]. Xin et al. reported that GBM-derived ECs can be detected in 46.9% of clinical samples, and EC markers were up-regulated in GSC cells, indicating that GSC might trans-differentiate into EC and vice versa [74].

### 7.2. Immune Niche of GSC

The immune niche of GBM consists of variance types of immune cells, among which macrophages are the most abundant cell types. Macrophages can be divided into M1 and M2 phenotypes [75]. The M1 subtype has the effect of killing tumor cells while the M2 subtype facilitates tumor survival by suppressing adaptive immunity of Th1 cells. Glioma-infiltrating myeloid cell (GIM) belongs to the M2 phenotype, and promotes immune-suppression of GBM and survival of GSC [76]. Tumor associated macrophages (TAMs) are immunosuppressive cells in GBMs, positively associated with tumor malignancy and negatively relating with patient survival [77]. Via releasing molecules such as macrophage colony-stimulating factor (M-CSF) and TGFβ, GSC is able to promote the M2-polarization of TAM [78] (Figure 4). In addition, GSC could recruit M2-polarized TAM to the hypoxic niche. Once microglia/macrophages are recruited by GSCs, they secrete TGFβ1 and IL10 to transfer into immunosuppressive cells [79]. The exact role of TAM in the immune niche of GSC demands further exploration. Chimeric antigen receptor T cell therapy of GBM has been a hot topic recently [80]. Effector T cells enhance sensitivity of GBM to lysis, which could be reduced by HIF1α-dependent NANOG. It was reported that GSCs suppress proliferation of effector T cells via secretion of hypoxia-related galectin-3 [81]. CD8^+^ cytotoxic T lymphocyte (CTL) could sensitize GBM to conventional therapies [82]. Under hypoxia, cytotoxity of CTL is impeded by hypoxia-induced interleukin-6-activated STAT3 pathways, leading to decreased survival of GBM patients [83]. Hypoxia also upregulates PD-L1 expression of activated T cells which suppress immunity of GBM.

### 7.3. Hypoxia/Necrotic Niche of GSC

Apart from the GSC immune niche, the hypoxia/necrotic niche is identified as another niche of GSC, in which hypoxia facilitates proliferation and self-renewal of GSC by both induction of stem cell signatures and upregulation of HIF1α and VEGF [84]. Histologically, pseudopalisading necrosis (PPN), the hypoxic area in GBM surrounded by numerous intensively packed cancer cells, is a fundamental histologic hallmark of GBM [85]. Hypoxia gives rise to acidification which then upregulates HIF expression and promotes GSC maintenance [86]. Pharmacological inhibition of HSP90, a hypoxia-regulated chaperone protein, could downregulate HIF expression and decrease oncogenicity of GBM [86]. In addition, hypoxia also has a great impact on the immune niche of GBM by promoting M2-like macrophage polarization and producing an immunosuppressive microenvironment [87,88].

### 7.4. ECM Niche of GSC

The extracellular matrix (ECM) niche is generally deemed as a section of the peri-arteriolar and peri-vascular microenvironment, although some reports indicate that it is an independent niche [89,90]. It is suggested that the ECM niche is established by GSC itself in that GSC could deposit components of the ECM niche. In addition, this GSC niche also consists of extracellular vesicles, glycoprotein and proteoglycan, laminin, and tenascin-C (TNC) [91,92] (Figure 5). The former three factors are generated by ECs. Laminin is secreted by GSC-related EC and facilitates GBM progression. TNC is an underlying GSC signature expressed by differentiated GBM cells [92]. Anti-TNC aptamers could inhibit GBM progression [92,93]. Very interestingly, by comprehensively studying the complicated molecular interacting networks of hypoxia regulated genes (HRGs-MINW) in GBM, Mao et al. found that CEBPD is a master transcriptional factor for the HRGs-MINW, and ECM mediated activation of EGFR/PI3K is a main down-stream pathway [94], which lends credence to the importance of the ECM niche for GBM. Notably, ECM proteins are critical hypoxia induced targets, and fibronectin (FN1), a key component of ECM, and integrin interaction mediated EGFR phosphorylation is a key step for CEBPD induced GBM progression [94]. As described above, integrin α6 is a potential GSC marker and plays and important role for the tumorigenicity of GSC [68,69].

### 7.5. Peri-Arteriolar Niche of GSC

Besides peri-vascular niche, peri-arteriolar area is another blood vessel related niche, but it has different features. Structurally, five layers from the outer rim to the lumen constitute the walls of the arterioles in the GSC peri-arteriolar niche: tunica adventitia, tunica elastica externa, tunica media, tunica elastica interna, and endothelium. Peri-arteriolar GSC niche resembles that of HSC in the bone marrow in that five of the same factors, SDF-1α, CXCR4, OPN, CD44, and CatK, are identified in both niches [95]. Notably, hypoxia is also a remarkable feature of the Peri-arteriolar niche in GBM [96].

### 7.6. Interactions between the Five GSC Niches

There are intimate interplays between the five GSC niches mentioned above. TAM could trans-differentiate into EC, indicating that EC constitutes part of GSC immune niche [97]. It is obvious that microglia/macrophage is overlapped in both immune niche and peri-vascular niche. VEGF is expressed in both GSC and TAM, suggesting the existence of a pro-angiogenesis signaling pathway in GSC immune niche [97]. Hypoxia often presents in the development and metastasis of lymphocytes while oxygen-deficiency damages the proliferation and secretion of cytotoxic T lymphocytes [98]. Hypoxia improves the lytic ability of CD8^+^ T cells and upregulates secretion of interferon-gamma by CD4^+^ T cells in vitro [98] (Figure 6). In addition, hypoxia could activate the pSTAT pathway to promote expression of immunosuppressive cytokines such as CCL2 and CFS1, both of which suppress proliferation of T-cells and promote infiltration of macrophages, facilitating tumor invasion and progression [99]. More specifically, HIF1α inhibits the differentiation of Foxp3^+^ Tregs by facilitating ubiquitination and degradation of Foxp3 in Th17 cells [100]. Meanwhile, the immune niche also has great impact on the HIF. Activated T cell receptor (TCR) enhances synthesis and stabilization of HIF1α in hypoxia. Moreover, CD4^+^ type1 Treg (Tr1), CD4^+^ T helper 17 (Th17), and CD8^+^ T cell in the immune niche of GBM could also stabilize HIF1α, implying a close association between the immune and hypoxic niches [101].

Formation of a GBM hypoxia/necrotic niche is possibly dependent on the peri-vascular niche, where tortuous vessels with insufficient perfusion are formed in the hypoxic necrosis areas in GBMs. Intriguingly, an upregulated level of HIF1α in GSCs results in enhancement of VEGF [102], which promotes malfunctional vessels. TAM could be activated by angiocrine-induced interleukin-6 (IL6) and subsequent argeinase-1 expression mediated by HIF2α, ultimately contributing to GBM progression [103]. Actually, peri-arteriolar niche, hypoxic niche, and immune niche might be the same type of GSC niche from three distinct viewpoints. Leukocyte-associated markers CD68, CD177, and MMP9 are expressed in the peri-arteriolar GSC niche where OPN is detected [104]. OPN co-localizes with CD68^+^ macrophages [104]. Peri-arteriolar niche is intimately associated with hypoxia; the possible reason is that arterioles are transport vessels rather than exchange vessels, thereby peri-hypoxic regions surrounding arterioles still form despite the oxygenated blood running along the arteriolar lumen. In addition, ECM niche is generally considered as part of peri-vascular niche, and plays essential roles in the interactions between GSC and immune niche [105].

All five GSC niches are close to necrotic regions where VEGF and HIF1α are overexpressed. Upregulated VEGF and HIF1α further induce expression of SDF1α and CXCR4, both being critical for the maintenance of GSC stemness [106]. TAM also stimulates expression of SDF1α and CXCR4. Given the striking amounts of cell types and proteins overlapped in the five niches, they might be complementary and integrated as a comprehensive niche: the hypoxic peri-arteriolar niche of GSC, resembling HSC niche in the bone marrow [96].

Disrupting interactions between GSC and its protective niches might enhance anti-GBM therapeutic sensitivity. CXCR4 is a significant factor involved in the GSC niche. Lee et al. conducted a phase I clinical trial and demonstrated the safety of plerixafor, a reversible CXCR4 inhibitor, plus bevacizumab strategy in GBM [107].

## 8. GSC and Hypoxia-Related Autophagy

Another remarkable hypoxia induced response of GBM to obtain therapy resistance and tumorigenicity is autophagy [87,108]. Autophagy is a highly-conserved catabolic reaction during evolution, which is a downstream event of mTOR hyper-activation. When oxygen is sufficient, degradation of HIF1α lead to activation of mammalian target of rapamycin (mTOR) and inhibition of autophagy. Conversely, under hypoxia, autophagy is induced through abnormal activation of Notch, Wnt/β-catenin, Hedgehog signaling pathways, and autophagy-related 9 A (ATG9A) in GBM [109,110] (Figure 7). Autophagy modulates protein degradation and turnover of neuronal stem cells (NSCs). Upregulation of autophagy could promote self-renewal and expansion of GSCs. In hypoxic condition, autophagy is also closely related to dysregulated metabolism pathways in GSCs in that autophagy provides a source of energy for tumor cells [109]. Autophagy functions as a protective mechanism against chemotherapy in GBM. For instance, temozolomide (TMZ) resistance in GBM is partly attributed to induced autophagy. Fortunately, scientists have identified potential novel drugs targeting autophagy. Inhibition of autophagy enhances chemosensitivity of GSCs to TMZ by igniting ferroptosis [111]. Tocilizumab, an inhibitor of IL6 receptor, decreases autophagy and upregulates chemosensitivity of TMZ in GBM [112]. GBM patients treated with chloroquine (CQ), a kind of autophagy flux suppressant, display reduced chemoresistance and better survival [113]. Inhibitor of the MST4-ATG4B signaling axis suppresses autophagy, which then decreases the malignancy of GBM [114]. Taken together, autophagy increases hypoxia-induced chemoresistance of GBM while inhibitors of autophagy have the capacity to reverse this phenomenon, being a potential therapeutic target for GBM.

## 9. GSC and Hypoxia-Related Therapeutic Resistance

One of the essential reasons underlying the dismal prognosis of GBM patients is the intrinsic therapy-resistance feature of GBM cells. Despite the combination of surgical resection, chemo-and-radiotherapy, prognoses of GBM patients remain unfavorable with the median survival span around 14–16 months [115]. Actually, the majority of GBM patients show inevitable recurrence. Potential causes of therapeutic resistance in GBM need urgent investigation (Table 2). Due to the ability of GSCs to infiltrate proximate normal tissues and elevated levels of tumor vascularization, it is difficult to perform complete surgical resection for GBMs. Remanent tumor cells at the margin of post-surgery are more proliferative and aggressive. Frequent exposure to irradiation and subsequent activation of DNA-damage response resulted in alterations in cell cycle and cell cycle-related proteins, enhanced expression of Notch pathway, and production of insulin-like growth factor 1 (IGF1) in GSC, all of which at least contribute to GBM radio resistance [116]. TMZ-resistance in GBM correlates with increased levels of DNA double-strand break and p38-ERK1/2 axis [117]. There are other mechanisms of chemo-resistance in GBM, such as upregulated activation of COX2 [118] and elevated expression of multidrug resistance-associated protein 1 (MRP1) transporter in GSCs which could expel chemo-therapeutic drugs to extracellular medium [119].

## 10. GSC and Hypoxia-Related Chemotherapy

At present, TMZ is the only first-line effective chemo-agent for GBM, despite the plentiful clinical trials on chemotherapeutic agents currently under way. Interestingly, the efficacy of TMZ associates with activity of HIF1α [120]. Recent studies validated that combining TMZ with other molecules has clinical efficacy (Table 3). It has been widely recognized that promotor methylation of O6-methylguanine-DNA-methyltransferase (MGMT), a DNA repair enzyme, is a reliable marker for TMZ sensitivity of GBM treatment. There are other molecular mechanisms that can affect the TMZ sensitivity. Activation of epidermal growth factor receptor variant III (EGFRvIII) could enhance hypoxia-induced death in GBM [121]. Pretreatment with S-nitroso-N-acetylpenicillamine (SNAP) in GBM patients induces expression of HIF1α [122]. Based on these findings, recent studies revealed that TMZ, combined with either EGFRvIII or SNAP, could significantly prolong survival of patients with MGMT promoter methylated GBM [117]. Metformin (MET) is commonly utilized as an antidiabetic agent. It was suggested that TMZ plus MET could revert chemoresistance in hypoxic condition via suppression of the PI3K/mTOR pathway in GBM [37]. Bevacizumab plus biweekly temozolomide is well tolerated in recurrent GBM patients [123]. However, aberrant vasculature could increase post-bevacizumab regional hypoxia in refractory GBM patients [124]. In addition, several medicines or agents can influence the efficacy of TMZ. Imipramine, an anti-depressant agent, could stimulate phenotypical switch from GSCs to non-GSCs in hypoxia [125], and TMZ plus either imipramine or tranylcypromine, another anti-depressant, could reduce the cytotoxic effect of TMZ under hypoxia [126]. Decitabine (DAC), a DNA hypomethylating agent, could increase the cytotoxicity of TMZ in GBM [127]. N45, a kind of steroidal saponin with anti-neoplasm efficacy, could inhibit cellular proliferation through the hypoxia-associated ROS/PI3K/Akt pathway in TMZ-resistant GBM [128].

In addition to TMZ, there are multiple chemo-agents undergoing research. Tacrolimus (FK506) has capacity to increase chemosensitivity of GSC and reduce GBM tumor volume and hypoxia-induced surface markers (ki67, GFAP and nestin) in GSC [119]. Bortezomib (BTZ) could stabilize expression of HIF1α in a mice model [129], and Ursodeoxycholic acid (UDCA) combined with BTZ has a synergistic effect on treatment of GBM [130]. BAL101553 is an effective chemo-agent in targeting hypoxia-mediated angiogenesis of GBM, and EB1 might be a response-predictive marker of BAL101553 treatment [131]. Although bevacizumab attributes to regional hypoxia in recurrent GBM patients, the addition of anti-VEGF antibody bevacizumab to carmustine would not enhance incidence of hematologic toxicity, validating the safety of this combinatory therapy in treating GBM patients [132]. Evofosfamide (TH-302) plus bevacizumab (Bev) strategy is well tolerated in Bev-refractory GBM patients [133]. Here, Evofosfamide is activated in hypoxia to obtain the capacity of discharging the DNA-damaging Br-IPM (bromo-isophosphoramide mustard) moiety [134]. As an oncogenic driver of GBM, the expression of EGFR is upregulated during hypoxia, and the anti-EGFR antibody nimotuzumab exhibited beneficial effects for the survival of GBM patients [135]. Targeting HIFs is a promising chemo-therapeutic strategy in treating GBM. One issue to bear in mind is that strategies should be tailored to inhibit HIFs in a specific pattern while leaving non-neoplastic cells unaffected. Two categories of digitalis exhibited potential effects in treating GBM: digoxin and digitoxin. Digoxin is a heart glycoside involved in translational inhibition of both HIF1α and HIF2α [136]. It was identified that HIF-related digoxin is effective in targeting GBM under hypoxia. Digitoxin is a cardiac glycoside and a suppressant of HIF1α, which can target GSCs with high specificity [137]. In addition, Cetuximab and Topotecan have the potential to target GBM by reducing translation of HIF1α [138,139].

**Table 3 cancers-15-02613-t003:** Advances in hypoxia-related chemotherapy targeting GSC.

Chemo Agent	Function	Reference
TMZ	associates with HIF-1α and prolong survival span of GBM patients	(Lo Dico et al., 2018 [120]; Struve et al., 2020 [117])
TMZ plus EGFRvIII	EGFRvIII enhances hypoxia-induced death and cooperates with TMZ to prolong survival of patients with MGMT promoter methylated GBM	(Struve et al., 2020 [117]; Luger et al., 2020 [121])
TMZ plus SNAP	SNAP induces HIF-1α and cooperates with TMZ to benefit survival span of GBM patients with MGMT promoter methylated	(Tsai et al., 2019 [122])
TMZ plus metformin	reverts chemoresistance of GBM during hypoxia via inhibition of PI3K/mTOR pathway	(Lo Dico et al., 2019 [120])
Biweekly TMZ plus bevacizumab	well tolerated by refractory GBM patients but increases regional hypoxia	(Badruddoja et al., 2017 [123]; Gerstner et al., 2020 [124])
TMZ plus Decitabine	increases cytotoxicity of HIF-1α-related chemo-agent	(Gallitto et al., 2020 [127])
TMZ plus imipramine	reduces cytotoxic effect of TMZ under hypoxia	(Bielecka and Obuchowicz, 2017 [126])
TMZ plus tranylcypromine	reduces cytotoxic effect of TMZ under hypoxia	(Bielecka and Obuchowicz, 2017 [126])
N45	inhibits proliferation through hypoxia-associated ROS/PI3K/Akt pathway in TMZ-resistant GBM	(Zhang et al., 2020 [128])
Tacrolimus (FK506)	reduce GBM tumor volume and hypoxia-induce surface markers (ki67, GFAP and nestin) in GSC	(Torres et al., 2018 [119])
UDCA bortezomib plus BTZ	stabilizes expression of HIF-1α and a promising therapy for GBM patients	(Yao et al., 2020 [130])
BAL101553	targets hypoxia-mediated angiogenesis of GBM	(Bergès et al., 2020 [131])
Bevacizumab plus carmustine	not enhance incidence of hematologic toxicity but attributes to regional hypoxia in recurrent GBM	(Yerram et al., 2019 [132])
nimotuzumab	an anti-EGFR antibody that upregulates survival span of GBM patients	(Ronellenfitsch et al., 2018 [135])
Evofosfamide plus bevacizumab	activated during hypoxia and well tolerated by bevacizumab-regressive GBM patients	(Brenner et al., 2018 [133]; Takakusagi et al., 2018 [134])
amitriptyline	stimulates phenotypical switch from GSCs to non-GSCs	(Bielecka-Wajdman et al., 2017 [125])
digoxin	inhibits HIF-1α and HIF-2α to target GBM	(Patocka et al., 2020 [136])
digitoxin	suppresses HIF-1α to target GSCs	(Lee et al., 2017 [137])
Cetuximab	reduces translation of HIF-1α to target GBM	(Ferreira et al., 2020 [138])
Topotecan	reduces translation of HIF-1α to target GBM	(Bernstock et al., 2017 [139])

## 11. GSC and Hypoxia-Related Radiotherapy

Apart from chemotherapy, radiotherapy of GBM also obtained breakthroughs in recent years (Table 4), and a list of chemical agents or drugs are identified to have beneficial effects on radiotherapy. The flavonoid extracted from Eucommia ulmoides can increase the effect of GBM radiotherapy by downregulating the HIFa/MMP-2 pathway and inducing apoptosis during radiotherapy [140]. Olaparib is identified as a promising radiosensitizer that improves prognosis of GBM patients. A multicenter clinical trial revealed that Olaparib plus temozolomide and intensity modulated radiotherapy could improve patient prognosis while sparing healthy tissues and preserving neurocognitive functions in GBM patients [141].

## 12. GSC and Hypoxia-Related Radio-, Immunotherapy

Immune therapy is one of the most promising ones for cancer, and has achieved great advances in several kinds of cancer [143]. However, to date, immune therapy is not successful in GBM treatment [82]. Plenty of studies were performed to find immunotherapeutic targeting of GSCs. GSCs secrete periostin (POSTN) to recruit cancer-supportive M2 phenotype of TAMs, which facilitates formation of immunosuppressive niche. Immune checkpoint inhibitor represents a kind of promising immunotherapy strategy in several kinds of tumors. Utilizing PD-1 inhibitors (nivolumab or pembrolizumab), a study indicated that therapeutic responses of GBM patients to anti-PD-1 immunotherapy correlate with specific molecular alterations such as PTEN mutation and MAPK enrichment. In general, manipulation of macrophage type between M1 and M2, and targeting immune checkpoint molecules, might act as a potential novel immunosuppressive strategy for GBM [35,75]. Targeting the microenvironment of GSCs with nanoparticles was reported to be effective in GBM immunotherapy. Distinct from the conventional delivery method, unique properties of nanoparticles enable successful penetration of drugs into GBM niche [142]. Nanoparticles could also be combined with chemo-radiotherapy, and photodynamic therapy. In the future, more individualized nanoplatforms ought to be designed to suit TME at distinct developing stages of GBM [142]. As described above, hypoxia plays important roles in regulating the immune niche of GBM [79,87,144], and might be promising therapeutic targets for immunotherapy.

## 13. GSC and Hyperbaric Oxygen Therapy

As a potential approach to reverse the hypoxic microenvironment in GBM, hyperbaric oxygen therapy (HBOT) is an innovative and effective adjuvant therapy to chemotherapy and irradiation in post-surgical GBM patients [145,146] (Table 5). HBOT enhances oxygen pressure in intratumoral, peritumoral, vascular tissues, and mitochondrial organelles, thus raising radio- and chemosensitivity of GBM cells [147]. Currently, there are two kinds of HBOT-radiotherapies: radiotherapy during HBOT; and radiation within 15 min after HBOT. Performing radiotherapy and HBOT simultaneously could prolong survival span of GBM patients. Only a small fraction of patients demonstrated severe side-effects such as conclusive seizure and radiation-correlated necrosis [148]. Nonetheless, conducting radiation during HBOT has not been used as a standard therapeutic modality, and potential reasons are: difficulties for radiation establishment; and underlying damage to surrounding normal tissues [149]. The rationale for performing radiation within 15 min subsequent to HBOT is that radiosensitivity of GBM peaks at exactly this time point. In addition, irradiation combined with chemical agents performed 15 min after HBOT caused no late toxicities in GBM patients [150]. A study indicated that performing radiation after HBOT could improve prognoses of GBM patients, with the 2-year overall survival (OS) and progression-free survival (PFS) rates reaching to 46.5% and 35.4%, respectively [151].

## 14. Conclusions

GBM is recognized as one of the most dismal brain tumors with unfavorable prognosis despite surgical resection, radio- and chemotherapy. GSCs are a small proportion of GBM cells which exhibit stem-like features such as self-renewal, invasion, and recapitulating the parent tumor, being major causes of GBM resistance. Hypoxia, tumor niches, and autophagy contribute to the maintenance and amplification of GSCs. Hypoxia plays a significant role, mainly mediated by HIF, in tumorigenesis of GBM including self-renewal of GSC, neovascularization, metabolism, IDH-mutation, Notch signaling pathway, and radio- and chemoresistance. Besides GSCs, the surrounding niches of GSCs also promote malignancy of GBM. Five niches are elucidated in this review: immune niche, peri-vascular niche, hypoxia/necrotic niche, peri-arteriolar niche, and extracellular matrix (ECM) niche. Interestingly, these five niches intimately interact with each other and might be integrated into a comprehensive category of niche: the hypoxic peri-arteriolar GSC niche. Autophagy, which can be boosted by hypoxia, is another protective mechanism against chemotherapy and is a potential therapeutic target for GBM. Other chemotherapeutic drugs, and novel adjuvant therapies such as HBOT, that can increase the effects of chemo- or radio-therapy and immunotherapy are also discussed in this paper. However, despite accumulated advantages, an effective treatment targeting hypoxia and GSCs is still lacking, mainly due to the fact that a lot of genes are induced by hypoxia which formed a complicated molecular interacting network affecting many biological processes [144]. Therefore, detailed mechanisms underlying hypoxia induced responses of GSC should be explored. The present paper summarized key features relevant to the GSC and hypoxia. Based on comprehensive understanding of these progresses, further laboratory work and clinical trials on hypoxia and GSCs can be developed to better prolong the survival span of GBM patients.

## Figures and Tables

**Figure 1 cancers-15-02613-f001:**
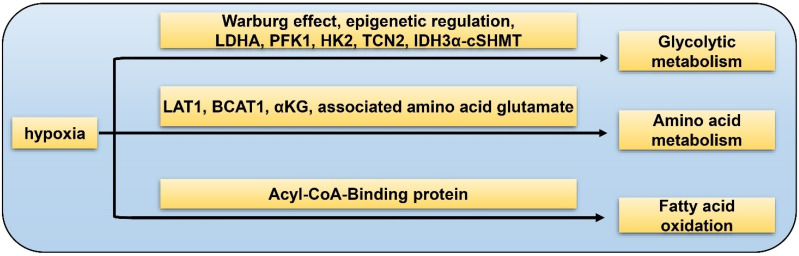
Hypoxia-related metabolism of GSC. Glycolytic metabolism: hypoxia promotes GSC glycolysis via Warburg effect, epigenetic regulation, LDHA, PFK1, HK2, TCN2, and IDH3α-cSHMT pathway; amino acid metabolism: hypoxia facilitates GSC glutamine metabolism by LAT1, BCAT1, αKG, and associated amino acid glutamate; fatty acid oxidation: hypoxia accelerates fatty acid metabolism through the Acyl-CoA-Binding protein.

**Figure 2 cancers-15-02613-f002:**
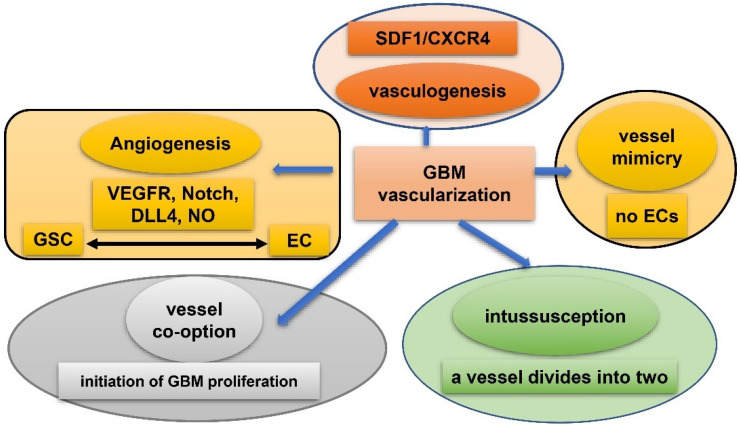
Five mechanisms of vascularization in glioblastoma. Angiogenesis: mainly attributed to interplay between GSCs and endothelial cells via VEGFR, Notch, DLL-4, and NO signals; vasculogenesis: bone marrow-derived endothelial progenitor cell (EPC) is recruited via SDF-1/CXCR-4 axis; vascular mimicry: tumor cells constitute vascular channels which have no endothelial cells but are still capable of transporting erythrocytes; vessel co-option: initiates proliferation of GBMs and is followed by angiogenesis; Intussusception: a blood vessel divides into two.

**Figure 3 cancers-15-02613-f003:**
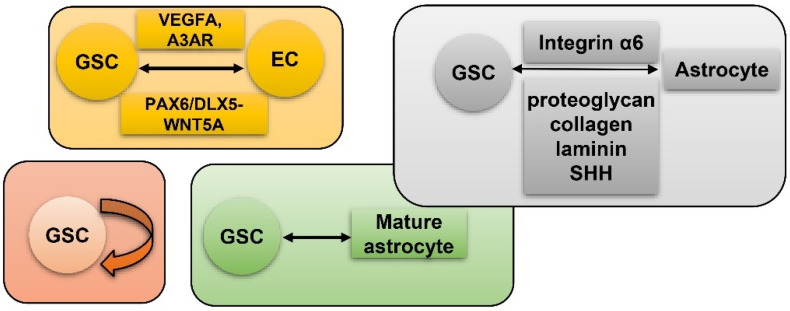
GSC and hypoxia-related vascularization. GSCs generate VEGFA and A_3_ AR to facilitate proliferation of ECs. Meanwhile, GSC and EC interact via the PAX6/DLX5-WNT5A axis; astrocytes generate proteoglycan, collagen, laminin, and SHH to interact with GSC-produced Integrin α6; self-renewal of GSC; mature GBM cell could de-differentiate into GSC.

**Figure 4 cancers-15-02613-f004:**
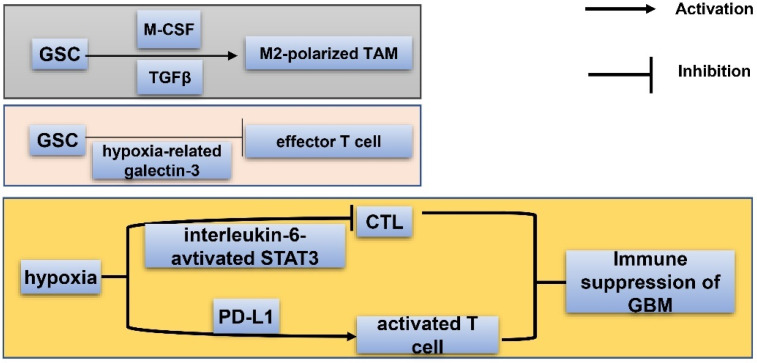
Immune niche of GSC. Via releasing macrophage colony-stimulating factor (M-CSF) and TGF-β, GSC could promote M2-polarization of TAM; by secreting hypoxia-related galectin-3, GSCs suppress proliferation of effector T cells; hypoxia enhances the PD-L1 level of activated T cells and induces interleukin-6-activated STAT3 to impede cytotoxity of CTLs (cytotoxic T lymphocytes). The above described processes all contribute to immune suppression and decreased survival of GBM patients.

**Figure 5 cancers-15-02613-f005:**
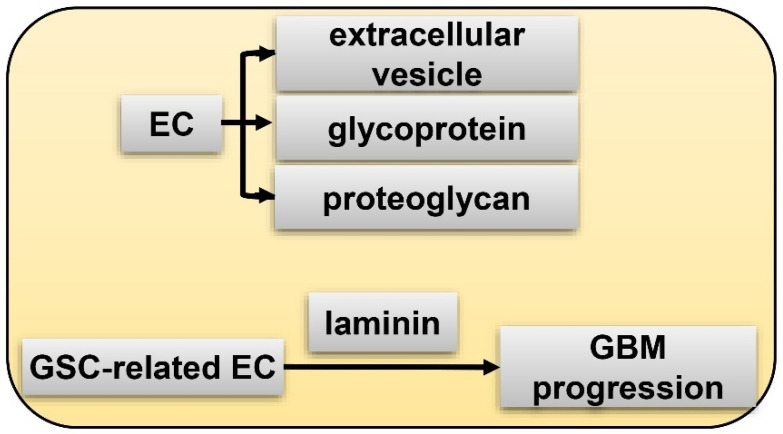
ECM (extracellular matrix) niche of GSC. EC generates extracellular vesicle, glycoprotein, and proteoglycan; GSC-related EC secretes laminin to facilitate GBM progression.

**Figure 6 cancers-15-02613-f006:**
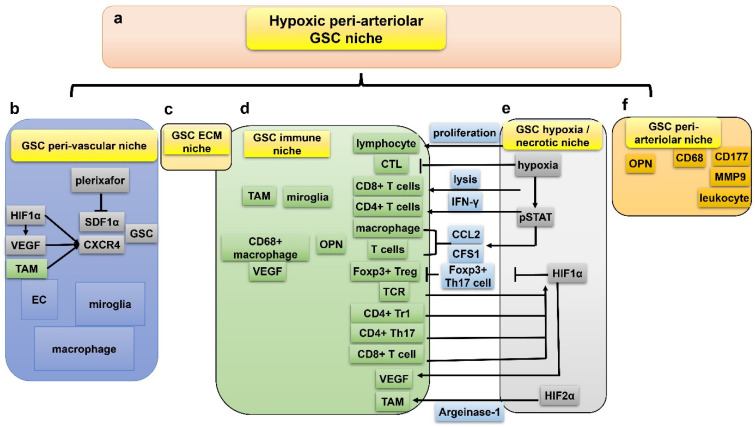
Five hypoxia-related GSC niches are closely associated and might be integrated into the hypoxic peri-arteriolar niche. (**a**) The integrated hypoxic peri-arteriolar niche is a combined concept from the inter-played 5 niches (**b**–**f**); (**b**) iInteraction between peri-vascular niche and hypoxia/necrotic niche via VEGF, SDF1α, CXCR4; (**c**) ECM niche (see details in Figure 5); (**d**) interaction between immune niche and hypoxia/necrotic niche via HIF1α, HIF2α, lymphocytes, CD8^+^ T cells, interferon-γ, CD4^+^ T cells, pSTAT pathway, CCL2, CFS1, macrophages, Foxp3^+^ Tregs, Th17 cells, TCR, CD4^+^ type1 Treg, and CD4^+^ T helper 17; (**e**) GSC hypoxia/necrotic niche; (**f**) interaction between immune niche and peri-arteriolar niche via CD177, MMP9, OPN, and CD68^+^ macrophages.

**Figure 7 cancers-15-02613-f007:**
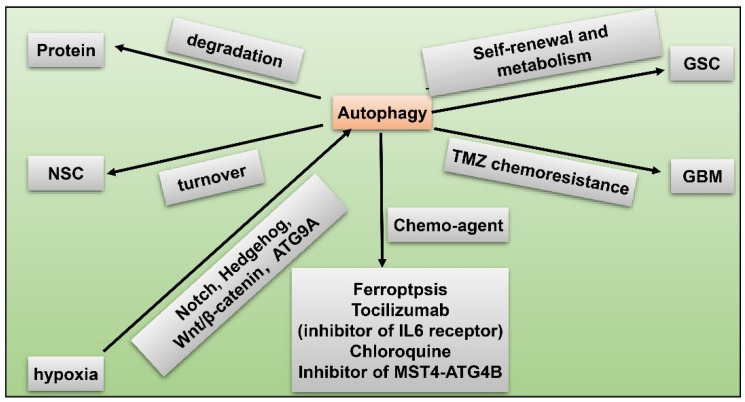
GSC and hypoxia-related autophagy. Hypoxia induces autophagy via activated Notch, Hedgehog, Wnt/β-catenin pathways, and autophagy-related 9 A (ATG9A) in GBM; autophagy modulates protein degradation; autophagy regulates turnover of NSCs; upregulated autophagy promotes self-renewal and metabolism of GSC; autophagy facilitates TMZ resistance in GBM; chemo-agents such as ferroptosis, tocilizumab, chloroquine, and inhibitor of MST4-ATG4B axis suppress autophagy and reduce malignancy of GBM.

**Table 1 cancers-15-02613-t001:** GSC and hypoxia-regulated signatures, genes, and pathways.

Signature	Gene	Pathways	lncRNA	Protein
CD9	EGFR	DLK1	lncRNA H19	HILPDA (HIG2)
SP	TP53 mutation	Notch (CBF1)		
CD133	IDH mutation	VEGF		
Olig2	MCT4	JAK1//2-STAT3		
integrin αβ	PP2A	Wnt (TCF-1, LEF-1)		
ALDH	Klf4	avβ8-integrin-TGF-β1		
CD44	ABCB1			
Sox2	PTEN			
Oct4	PML			
nestin				

**Table 2 cancers-15-02613-t002:** Causes of hypoxia-related therapeutic resistance in GSC.

Post-Surgical Recurrence	Radio Resistance	Chemo Resistance
GSC infiltrate proximate normal tissues	cell cycles alter	DNA double-strand break upregulates
tumor vascularization upregulates	cell cycle-related proteins alter	p38-ERK1/2 axis increases
diffusion around proximate tissues	expression of Notch increases	COX2 elevates
	GSC produces insulin-like growth factor 1 (IGF1)	multidrug resistance-associated protein 1 (MRP1)
	DNA-damage response activates via musashi-1	

**Table 4 cancers-15-02613-t004:** Advances in hypoxia-related radio-and-immune therapy targeting GSC.

Agent	Mechanism	Function	Reference
total flavonoid of *Eucommia ulmoides*	downregulates HIF-a/MMP-2 pathway and upregulates apoptosis	increase effect of GBM radiotherapy	(Wang et al., 2019 [140])
Olaparib	a promising radiosensitizer	improves prognosis of GBM patients	(Lesueur et al., 2019 [141])
Olaparib plus temozolomide	combined with intensity modulated radiotherapy	spares healthy tissues and preserves neurocognitive functions to improve prognosis of GBM patients	(Lesueur et al., 2019 [141])
nivolumab	a PD-1 inhibitor associates with PTEN mutation and MAPK enrichment	displays therapeutic efficacy of GBM	(Zhao et al., 2019 [35])
pembrolizumab	a PD-1 inhibitor associates with PTEN mutation and MAPK enrichment	displays therapeutic efficacy of GBM	(Hsu et al., 2020 [75])
nanoparticles	penetrates GBM niche and combines with chemo-, radio- and photodynamic therapies	displays therapeutic efficacy of GBM	(Yang et al., 2021 [142])

**Table 5 cancers-15-02613-t005:** Hyperbaric oxygen therapy (HBOT) in GSC.

Definition	Category	Benefit	Side-Effect	Difficulty	Reference
An adjuvant therapy to chemo-and-radio therapy in post-surgical GBM patients	radiotherapy during HBOT	prolong survival span of GBM patients	conclusive seizure and radiation-correlated necrosis	radiation establishment and underlying damage to normal tissues surrounded	(Chang, 1977 [148]; Ogawa et al., 2013 [149])
	radiation within 15 min after HBOT	improve prognoses of GBM patients, with progression-free survival rate reaching 46.5%	cause no late toxicities in GBM	requires more clinical validation	(Ogawa et al., 2012 [150]; Yahara et al., 2017 [151])

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
