# Peer review of "The Role of Hypoxia and Cancer Stem Cells in Development of Glioblastoma"

_cancers, 2023, doi:10.3390/cancers15092613_

Round 1

Reviewer 1 Report

The authors propose for publication a review titled «Glioblastoma and Cancer Stem Cells with Hypoxia». The text of the review is written correctly, without significant errors. The main problem of the work is the lack of newly drawn conclusions; in essence, the work is a simple retelling of known data. The abstract does not contain information about the contribution of the authors to the work, does not describe new ideas or conclusions proposed by them. 

The introduction to articles of the review type should end with a brief statement of the problem, which the author must solve by analyzing existing data and generating new conclusions based on them. In this article, such a formulation of the problem is absent.

The conclusion of this review does not contain specific recommendations for solving the identified problems, but in fact is a summary of the entire article.

The authors of the article also do not refer to a number of important publications. For example, the named concept of "five niches of glioblastoma stem cells" was first described in the article « Verhaak RG, Hoadley KA, Purdom E, Wang V, Qi Y, Wilkerson MD, et al.  Integrated genomic analysis identifies clinically relevant subtypes of glioblastoma characterized by abnormalities in PDGFRA, IDH1, EGFR, and NF1. Cancer Cell 2010; 17: 98–110», reference to which does not appear in the text.

More importantly, previously published work «Boyd NH, Tran AN, Bernstock JD, Etminan T, Jones AB, Gillespie GY, Friedman GK, Hjelmeland AB. Glioma stem cells and their roles within the hypoxic tumor microenvironment. Theranostics. 2021 Jan 1;11(2):665-683. doi: 10.7150/thno.41692. PMID: 33391498; PMCID: PMC7738846» already describes the data presented in many ways and generalizes them.

Little attention is paid in the text of the article to the role of the extracellular matrix in influencing GSCs.

Based on the collected information, it is necessary to generate new ideas and draw new conclusions. Each subchapter should be supplemented by a paragraph containing conclusions from the information received, representing previously unidentified correlations, connections, features, etc.

Abstract, introduction, conclusion should also be supplemented in accordance with the recommendations above.

Author Response

Reviewer 1

  1. The authors propose for publication a review titled «Glioblastoma and Cancer Stem Cells with Hypoxia». The text of the review is written correctly, without significant errors. The main problem of the work is the lack of newly drawn conclusions; in essence, the work is a simple retelling of known data. The abstract does not contain information about the contribution of the authors to the work, does not describe new ideas or conclusions proposed by them.

Answer: Thank you for the suggestion. We substantially revised the abstract and performed more discussion and subsequent conclusions to enhance the quality of the review.

  1. The introduction to articles of the review type should end with a brief statement of the problem, which the author must solve by analyzing existing data and generating new conclusions based on them. In this article, such a formulation of the problem is absent.

Answer: Thank you for the suggestion. We revised the structure of the whole manuscript, including the introduction part, where we raised the problem and potential solve approaches: “Upregulated HIF expression in hypoxia promotes proliferation, infiltration and self-renewal of GSC, ultimately leading to enhanced level of therapeutic-resistance. However, the relationship between hypoxia and GSCs in development of GBM is not clearly elaborated. Therefore, in our review, we recapitulate general features of GBM, describe GSC-related features, and delineate interaction between GSC and hypoxia. Given the importance of hypoxia for the initiation and progression of GBM and GSCs, comprehensive study and discussion of these issues would give us more insights into the biological features of GBM and provide novel avenues to develop promising treatments for GBM targeting hypoxia and GSC.”

  1. The conclusion of this review does not contain specific recommendations for solving the identified problems, but in fact is a summary of the entire article.

Answer: Thank you for the suggestion. We revised the structure of the whole manuscript, including the conclusion part as follows:

“GBM is recognized as one of the most dismal brain tumors with unfavorable prognosis despite surgical resection, radio-and chemotherapy. … However, despite accumulated advantages, effective treatment targeting hypoxia and GSCs is still lack, mainly due to the fact that a lot of genes are induced by hypoxia which formed a complicated molecular interacting network affecting many biological processes[143]. Therefore, detailed mechanisms underlying hypoxia induced responses of GSC should be explored. The present paper summarized key features relevant to the GSC and hypoxia. Based on comprehensive understanding of these progresses, further laboratory work and clinical trials on hypoxia and GSCs can be developed to better prolong the survival span of GBM patients.”

  1. The authors of the article also do not refer to a number of important publications. For example, the named concept of "five niches of glioblastoma stem cells" was first described in the article « Verhaak RG, Hoadley KA, Purdom E, Wang V, Qi Y, Wilkerson MD, et al. Integrated genomic analysis identifies clinically relevant subtypes of glioblastoma characterized by abnormalities in PDGFRA, IDH1, EGFR, and NF1. Cancer Cell 2010; 17: 98–110», reference to which does not appear in the text.

Answer: Thank you for the suggestion. We added this and several other relevant important publications.

  1. More importantly, previously published work «Boyd NH, Tran AN, Bernstock JD, Etminan T, Jones AB, Gillespie GY, Friedman GK, Hjelmeland AB. Glioma stem cells and their roles within the hypoxic tumor microenvironment. Theranostics. 2021 Jan 1;11(2):665-683. doi: 10.7150/thno.41692. PMID: 33391498; PMCID: PMC7738846» already describes the data presented in many ways and generalizes them.

Answer: Thank you for the suggestion. We studied this publication carefully, and added it as a reference. As a hot research areas, there might be certain overlap inevitably, but we presented in different angles and included latest progressions.

  1. Little attention is paid in the text of the article to the role of the extracellular matrix in influencing GSCs.

Answer: Thank you for the suggestion. We added more recent advantages in the extracellular matrix part and in relevant sections involving ECM. For example in the section “ECM niche of GSC”:”Very interestingly, by comprehensively studying the complicated molecular interacting networks of hypoxia regulated genes (HRGs-MINW) in GBM, Mao et al found that CEBPD is a master transcriptional factor for the HRGs-MINW, and ECM mediated ac-tivation of EGFR/PI3K is a main down-stream pathway[94], which lends credence to the importance of ECM niche for GBM. Notably, ECM proteins are critical hypoxia induced targets, and fibronectin (FN1), a key component of ECM, and integrin inter-action mediated EGFR phosphorylation is a key step for CEBPD induced GBM pro-gression [94]. Besides, as described above, integrin α6 is a potential GSC marker and plays important roles for the tumorigenicity of GSC [69,70]”

  1. Based on the collected information, it is necessary to generate new ideas and draw new conclusions. Each subchapter should be supplemented by a paragraph containing conclusions from the information received, representing previously unidentified correlations, connections, features, etc. Abstract, introduction, conclusion should also be supplemented in accordance with the recommendations above.

Answer: Thank you for the suggestion. we substantially revised the whole manuscript and added necessary conclusions in each paragraph. In addition, new ideas and new conclusions were summarized as described in the above answers.

Reviewer 2 Report

The authors reviewed roles of hypoxia in glioma stem cells (GSCs), consisting of a small subpopulation of glioblastoma (GBM) and causing worse prognosis by presenting a resistance to radio-chemotherapy and producing early recurrence of GBM. They summarized hypoxia-induced signatures, genes, signal pathways of GBM and demonstrated five GSC niches are integrated into one concept, peri-arteriolar niche. In addition, they showed that tumor-promoting autophagy in GBM is closely associated with hypoxia. They also described therapeutic strategy, including hyperbaric oxygen therapy, based on the GSCs’ behavior under hypoxia.

The manuscript extensively includes recent studies concerning relations between GSCs and hypoxic microenvironment to find a cue for effective treatment methods in GBM that can be translated to clinical practice. The contents of the text are not required correction or replenishment, but there are big problems in presenting Figures including a Table.    

1.     Figure 1. The legend is not properly presented the contents of the figure. In the legend, five items from (a) to (e) are described, but there are only three (a, b, c) are presented. Also, the figure and the legend are different contents.

2.     Figure 2. The same mistake as Fig.1. Contents in the figure are quite different from the figure legend.  

3.     Figure 4. Same mistake as above two figures.

4.     Figure 5. The figure is very complicated. The authors should present more simply the relation maps for five niches of GSC.

5.     Figure 6. Figure and the legend are not coincided.

6.     Table 1. These clinical features are thought to be unnecessary in this review article. The author had better omit the Table 1. 

Author Response

The authors reviewed roles of hypoxia in glioma stem cells (GSCs), consisting of a small subpopulation of glioblastoma (GBM) and causing worse prognosis by presenting a resistance to radio-chemotherapy and producing early recurrence of GBM. They summarized hypoxia-induced signatures, genes, signal pathways of GBM and demonstrated five GSC niches are integrated into one concept, peri-arteriolar niche. In addition, they showed that tumor-promoting autophagy in GBM is closely associated with hypoxia. They also described therapeutic strategy, including hyperbaric oxygen therapy, based on the GSCs’ behavior under hypoxia.

The manuscript extensively includes recent studies concerning relations between GSCs and hypoxic microenvironment to find a cue for effective treatment methods in GBM that can be translated to clinical practice. The contents of the text are not required correction or replenishment, but there are big problems in presenting Figures including a Table.   

  1. Figure 1. The legend is not properly presented the contents of the figure. In the legend, five items from (a) to (e) are described, but there are only three (a, b, c) are presented. Also, the figure and the legend are different contents.

Answer: Thank you for the suggestions. We corrected all the figure legends.

  1. Figure 2. The same mistake as Fig.1. Contents in the figure are quite different from the figure legend.

Answer: Thank you for the suggestions. We corrected all the figure legends.

  1. Figure 4. Same mistake as above two figures.

Answer: Thank you for the suggestions. We corrected all the figure legends.

  1. Figure 5. The figure is very complicated. The authors should present more simply the relation maps for five niches of GSC.

Answer: Thank you for the suggestions. We re-organized the figure in the revised version (Fig 6 in revised version) to make it more clear.

  1. Figure 6. Figure and the legend are not coincided.

Answer: Thank you for the suggestions. We corrected all the figure legends.

  1. Table 1. These clinical features are thought to be unnecessary in this review article. The author had better omit the Table 1.

Answer: Thank you for the suggestion. we also recognized that Table 1 is not necessary and omit it in the revised version.

Reviewer 3 Report

The manuscript "Glioblastoma and Cancer Stem Cells with Hypoxia" by Tingyu Shi and colleagues is a review of the literature concerning the importance of hypoxia and Hifs in glioblastoma biology and therapy. The review is rather complete and adjourned. However, the organization of the disparate findings reported in the text is often confusing. Moreover, grammar and style need a through revision. I listed below few examples of passages in the text that, in my opinion, need clarification.

The title is rather cryptic please find one title more explicative of the content  indicating the work as a review.

row 32 Please add a reference to the indicated incidence that appears lower than in many recent investigations.

rows 55-56 Hypoxia and Hif activation it is known to play a role in neuronal development but it is mainly restricted to neural crest cell migration and not on in the closure of neural tube. Please clarify and add the appropriate references.

row 257 "avtivated STAT3" please correct into activated STAT3.

Author Response

The manuscript "Glioblastoma and Cancer Stem Cells with Hypoxia" by Tingyu Shi and colleagues is a review of the literature concerning the importance of hypoxia and Hifs in glioblastoma biology and therapy. The review is rather complete and adjourned. However, the organization of the disparate findings reported in the text is often confusing. Moreover, grammar and style need a through revision. I listed below few examples of passages in the text that, in my opinion, need clarification.

  1. The title is rather cryptic please find one title more explicative of the content indicating the work as a review.

Answer: Thank you for the suggestion. We revised the title as “The role of hypoxia and cancer stem cells in development of Glioblastoma”

  1. row 32 Please add a reference to the indicated incidence that appears lower than in many recent investigations.

Answer: Thank you for the suggestion. we revised the data according to a more recent publish and added it as a reference.

  1. rows 55-56 Hypoxia and Hif activation it is known to play a role in neuronal development but it is mainly restricted to neural crest cell migration and not on in the closure of neural tube. Please clarify and add the appropriate references.

Answer: Thank you for the suggestion. We revised this and added relevant references.

  1. row 257 "avtivated STAT3" please correct into activated STAT3.

Answer: Thank you for the suggestion. we carefully revised the whole manuscript and corrected similar typo problems.

Round 2

Reviewer 3 Report

The revised manuscript was greatly improved by the authors. However, in my opinion, there are still minor revisions to be implemented in terms of English grammar and clarity of the style. The following are just few examples:

rr 14 Glioblastoma multiforme is more often used compared to Glioblastoma multiform, consider revising.

rr 72 "could exert antiproliferative efficacy" please correct e.g. could exert antiproliferative effects

rr 93 "One of the central issue" please correct "One of the central issues"

rr113-114 " are discovered in these four types respectively" please clarify